# The Emerging Roles and Clinical Potential of circSMARCA5 in Cancer

**DOI:** 10.3390/cells11193074

**Published:** 2022-09-30

**Authors:** Changning Xue, Jianxia Wei, Mengna Li, Shipeng Chen, Lemei Zheng, Yuting Zhan, Yumei Duan, Hongyu Deng, Wei Xiong, Guiyuan Li, Hui Li, Ming Zhou

**Affiliations:** 1NHC Key Laboratory of Carcinogenesis, Hunan Key Laboratory of Oncotarget Gene, Hunan Cancer Hospital, The Affiliated Cancer Hospital of Xiangya School of Medicine, Central South University, Changsha 410013, China; 2Cancer Research Institute, School of Basic Medical Sciences, Central South University, Changsha 410078, China; 3The Key Laboratory of Carcinogenesis and Cancer Invasion of the Chinese Ministry of Education, Central South University, Changsha 410078, China; 4The Second Xiangya Hospital, Central South University, Changsha 410011, China

**Keywords:** cancer, circSMARCA5, dual roles, ceRNA, tumor progression

## Abstract

Circular RNAs (circRNAs) are a type of endogenous non-coding RNA and a critical epigenetic regulation way that have a closed-loop structure and are highly stable, conserved, and tissue-specific, and they play an important role in the development of many diseases, including tumors, neurological diseases, and cardiovascular diseases. CircSMARCA5 is a circRNA formed by its parental gene SMARCA5 via back splicing which is dysregulated in expression in a variety of tumors and is involved in tumor development with dual functions as an oncogene or tumor suppressor. It not only serves as a competing endogenous RNA (ceRNA) by binding to various miRNAs, but it also interacts with RNA binding protein (RBP), regulating downstream gene expression; it also aids in DNA damage repair by regulating the transcription and expression of its parental gene. This review systematically summarized the expression and characteristics, dual biological functions, and molecular regulatory mechanisms of circSMARCA5 involved in carcinogenesis and tumor progression as well as the potential applications in early diagnosis and gene targeting therapy in tumors.

## 1. Introduction

Circular RNAs (circRNAs) are a type of non-coding RNA and a critical epigenetic regulation way that are formed by the back splicing of mRNA precursors (pre-mRNAs). CircRNAs range in length from 200 to 2000 bp, with the majority being around 500 bp. As circRNAs have a covalently closed loop structure without a 5′ cap and 3′ PolyA tail, they are not easily degraded by nucleic acid exonucleases, giving circRNAs greater stability than linear mRNAs [1,2,3]. CircRNAs have been linked to the development and progression of tumors [4,5,6,7,8] as well as a variety of diseases such as cardiovascular disease [9,10,11,12], neurological disease [13,14,15,16], autoimmune disease [17], and diabetes [18,19]. There are several main mechanisms of circRNA involved in the development and progression of diseases, including their function as miRNA sponges [20,21,22], interaction with RNA-binding proteins (RBPs) [23,24], effects on protein scaffolding effects [25,26,27], regulation of parental gene expression [27,28], and have peptide or protein-coding functions [29,30,31]. CircSMARCA5 (Circular RNA SMARCA5), also known as hsa-circ-0001445, is formed by the back splicing of exons 15 and 16 of its host gene, SMARCA5 (Figure 1), and has a genome length of 464 bp. The splice sequence is 269 bp in length, and its distribution is present in both the cytoplasm and nucleus [32,33,34]. SMARCA5 is an ISWI chromatin remodeling ATPase that belongs to the ATP-dependent chromatin remodeling complex SWI/SNF family [35,36], and is involved in tumor proliferation, invasion, chemoresistance, and progression, primarily through chromatin remodeling in DNA damage regions [37,38,39,40,41].

In this review, for the first time, we summarized the expression characteristics, biological functions, and potential mechanisms of circSMARCA5, as well as the potential applications in early diagnosis and gene targeting therapy in tumors, providing useful information and new insights into future research of circSMARCA5 and other circRNAs in tumors.

## 2. The Dual Functions of circSMARCA5 in Tumors

Many circRNAs have been shown to have dual functions in tumors, such as circFOXO3 [42,43] and CDR1as [44,45]. CircFOXO3 could promote the progression of gastric carcinoma, while it plays a tumor suppressive role in non-small cell lung cancer [42,43]. CDR1as was reported to promote colorectal cancer progression but suppress ovarian cancer progression [44,45]. Not only circRNAs, but also some protein-coding genes and other non-coding RNAs, such as bromodomain containing 7(BRD7) [46,47,48,49], YY1 [50,51], and miR-141 [50,52], play dual functions in tumor development and progression. According to recent research, circSMARCA5 is aberrantly expressed in a variety of tumors, has dual functions, and plays an important role in tumor progression via a complex network of gene regulation. CircSMARCA5 is down-regulated in most tumors, including breast cancer, glioblastoma, gastric cancer, multiple myeloma, colorectal cancer, liver cancer, non-small cell lung cancer, and cervical cancer, and plays a tumor suppressive role [32,34,53,54,55,56,57,58,59,60,61,62,63,64,65,66,67]. Whereas it is overexpressed in osteosarcoma, prostate cancer, and bladder cancer, where it plays a tumor-promoting role [33,68,69,70,71].Therefore, circSMARCA5 serves a tissue-specific dual role during tumor development and progression (Table 1).

### 2.1. CircSMARCA5 Exerts Anti-Tumor Function in a Variety of Tumors

#### 2.1.1. Liver Cancer

Liver cancer is a common group of malignancies, and primary liver cancer is the fifth most common cancer in the world [72]. Hepatocellular carcinoma (HCC) is the most common type of primary liver cancer, followed by Intrahepatic Cholangiocarcinoma (ICC), together accounting for more than 95% of primary liver malignancies [73]. In recent years, circSMARCA5 has been found to play role in the initiation and development of liver cancer [53,54,55,56,57]. Li et al., Zhang et al., Xu et al., as well as Yu et al. [53,54,55,56] all indicated that low expression of circSMARCA5 was detected in HCC tissues and cells. Overexpression of circSMARCA5 not only inhibited cell proliferation, migration, epithelial-mesenchymal transition, cell cycle, and invasive ability, but also promoted apoptosis [53,54,56]. Additionally, it has been demonstrated that circSMARCA5 was involved in glucose metabolism. Xu et al. [55] found that circSMARCA5 upregulation strikingly decreased ECAR and the level of glycolysis, glycolytic capacity, and lactate reserve in HCC cells. Furthermore, Yu et al. and Xu et al. [54,55] found the different regulatory pathways of circSMARCA5 in HCC. ExH-box helicase 9 (DHX9), a nuclear RNA decapping enzyme containing both dsRBD and RNA decapping enzyme structural domains [74], can reduce the expression of circSMARCA5 by its upregulation. The study by Yu et al. [54] showed that circSMARCA5 was determined to be a sponge of miR-17-3p and miR-181b-5p. According to previous reports, miR-17-3p and miR-181b-5p can promote HCC [75,76]. The growth and metastasis of HCC cells are inhibited by protecting TIMP metallopeptidase inhibitor 3 (TIMP3) from the downregulation of miR-17-3p and miR-181b-5p. Above all, the regulatory DHX9/circSMARCA5/miR-17-3p & miR-181b-5p/TIMP3 axis is an effective way for circSMARCA5 to exert its tumor suppressive effects [54]. Moreover, Xu et al. [55] carried out dual-luciferase assays, and pull-down assays which revealed that circSMARCA5 could bind to miR-942-5p and could significantly promote Arista-less-like homeobox 4 (ALX4) through targeting miR-942-5p. In another study, Lu et al. [57] showed that circSMARCA5 overexpression can significantly inhibit cell proliferation in ICC. Meanwhile, circSMARCA5 overexpression enhanced the chemosensitivity of ICC to cisplatin and gemcitabine. Therefore, the findings provide a possible therapeutic target for liver cancer.

#### 2.1.2. Multiple Myeloma

Multiple myeloma (MM) is a type of neoplastic plasma cell disease frequently seen in the aging population [77]. Liu et al. [58] indicated that low circSMARCA5 expression was associated with MM malignancy and poor prognosis. Their findings noted that circSMARCA5 was apparently decreased in MM cell lines and tissues, and overexpression of circSMARCA5 significantly inhibited cell proliferation and promoted apoptosis. Further experiments demonstrated that circSMARCA5 might act as a ceRNA to bind to miR-767-5p. Overall, all these data prove that circSMARCA5 can function as a tumor suppressor involved in the progression of MM as well as a great prognostic biomarker.

#### 2.1.3. Cervical Cancer

Cervical cancer (CC) is a prevalent gynecological malignancy, and high-risk subtypes of human papillomavirus (HPV) are the main cause of the disease [78,79]. Tian et al [34] found that circSMARCA5 expression level was significantly decreased in CC tissues and cell lines compared with normal controls. Moreover, overexpression of circSMARCA5 remarkedly inhibited cell proliferation, migration, and invasion of CC cell lines and induced cell cycle arrest, suggesting that circSMARCA5 exerts an inhibitory effect in cervical cancer. Mechanistic analyses proved that circSMARCA5 might serve as a ceRNA sponge to miR-620 and downregulate its expression [34]. These findings revealed that the circSMARCA5/miR-620 axis has a critical role in the development of CC, and circSMARCA5 may act as a potential prognostic biomarker as well as a target for therapeutic intervention in cervical cancer.

#### 2.1.4. Glioblastoma

Glioblastoma (GBM) is a malignant primary brain tumor with the highest incidence and is known to arise from glial progenitor cells. GBM patients have a poor prognosis [80]. Barbagallo et al [59,60] found that circSMARCA5 was significantly downregulated in GBM tissues, and its expression negatively correlated with the histological grade of GBM, indicating its negative role in malignant tumor progression, and overexpression of circSMARCA5 inhibited the migration of GBM cells [60]. They further revealed that circSMARCA5 could bind to the Serine and Arginine Rich Splicing Factor 1 (SRSF1) protein [60]. SRSF1 has been determined to be a splicing factor that is overexpressed in GBM and involved in the positive regulation of cell migration. CircSMARCA5 interacted with the splicing factor SRSF1 and regulated the SRSF1/serine and arginine-rich splicing factor 3(SRSF3)/PTB (PTB1 & PTB2) axis thus inhibiting the migration of GBM cell [60]. Furthermore, circSMARCA5 also regulated Vascular Endothelial Growth Factor A (VEGFA) through binding to SRSF1, inhibited angiogenesis by regulating the proportion of angiogenic and antiangiogenic vascular endothelial growth factor subtypes [59], thus acting as an anti-angiogenic molecule for the inhibition of tumor. Accordingly, these findings suggest that circSMARCA5 participates in GBM progression and functions as a therapeutic target.

#### 2.1.5. Breast Cancer

Breast cancer (BC) is a class of malignancies that occurs mostly in women and is the second leading cause of cancer death with a low survival rate [81]. The study by Xu et al. [32] indicated that circSMARCA5 was downregulated in BC tissues and BC cell lines, the expression of circSMARCA5 was significantly and negatively correlated with its parental gene, and the overexpression of circSMARCA5 promoted sensitivity to cisplatin treatment. Additionally, they proposed that circSMARCA5 inhibited the DNA damage repair function of BC by negatively regulating the level of linear SMARCA5 expression. CircSMARCA5 binds to its parental gene locus, forming an R-loop that suspended SMARCA5 transcription in its exon 15, leading to downregulation of SMARCA5 expression, and thus epigenetically resulting in the decreases of chromatin remodeling and DNA damage repair. These studies have shown that circSMARCA5 is a promising therapeutic target for breast cancer.

#### 2.1.6. Gastric Cancer

Gastric cancer (GC) is one of the most lethal cancers [82], and the 5-year overall survival rate of GC patients is less than 30% due to tumor metastasis and recurrence [83]. Both the studies by Cai et al. [61] and Li et al. [63] reported that low circSMARCA5 expression in GC cell lines and tissues was detected compared with the adjacent noncancerous tissues. The overexpression of circSMARCA5 could inhibit GC cell proliferation, migration, and invasion, and the glycolysis rate, glycolysis capacity, glucose uptake, and lactate production in the circSMARCA5-overexpresing group were significantly decreased compared to the control group [61,62]. Moreover, Cai et al. [61] demonstrated that a low expression level of circSMARCA5 was correlated to poorer overall survival and disease-free survival, and low circSMARCA5 expression was revealed as an independent unfavorable predictive factor for GC. In addition, further experiments confirmed that circSMARCA5 acts as a sponge for miR-346 in GC cell lines and upregulates the expression of F-box and leucine-rich repeat protein 2 (FBXL2) to promote GC progression [63]. FBXL2 is a highly conserved member of the F-box protein family, a component of the Skp-Cullin-F box (SCF) ubiquitin E3 ligase, which suppresses tumorigenesis by targeting and ubiquitinating oncoproteins [84]. Another study by Cai et al. [62] found that circSMARCA5 acted as a molecular sponge to inhibit the expression of miR-4295, and the miR-4295 could inhibit the expression of phosphatase and tensin homolog(PTEN) by binding with the 3′-UTR of PTEN mRNA. PTEN is an important metabolic factor which can negatively regulate PI3K/AKT signal transduction, reducing the glycolysis of tumor cells, thus inhibiting tumor progression [85]. These observations indicated circSMARCA5 may be a potential biomarker of therapeutic effects and diagnosis in GC.

#### 2.1.7. Non-Small Cell Lung Cancer

Lung cancer is one of the most common malignancies in the world and can be classified into small cell lung cancer (SCLC) and non-small cell lung cancer (NSCLC) according to its tumor cell morphology, of which NSCLC is the main form of lung cancer [86,87]. Wang et al. and Tong et al. [64,65] reported that circSMARCA5 expression was downregulated in both NSCLC cell lines and tissues, and overexpression of circSMARCA5 significantly inhibited the proliferation, migration, and invasive ability of NSCLC. Recombinant R10-homeobox A9 (HOXA9) protein was shown to inhibit epithelial-mesenchymal transition and reduce the invasion and migration of NSCLC cells [88]. Wang et al. [64] found that circSMARCA5 specifically could bind to miR-19b-3p, which inhibits HOXA9 gene expression. Therefore, circSMARCA5 can act as a tumor suppressor of NSCLC through the miR-19b-3p/HOXA9 axis. Moreover, the expression of circSMARCA5 was negatively correlated with tumor size, lymph node metastasis, and TNM stage, indicating that circSMARCA5 is negatively associated with disease progression [65]. Therefore, circSMARCA5 may represent a potential target for the diagnosis and prognosis of NSCLC patients.

#### 2.1.8. Colorectal Cancer

Colorectal cancer (CRC) is a malignant tumor of the gastrointestinal tract with high morbidity and mortality and is the fourth most lethal cancer in the world [89]. Miao et al. and Yang et al. [66,67] both revealed that the expression level of circSMARCA5 was significantly downregulated in CRC cell lines and tissues. The overexpression of circSMARCA5 could inhibit the proliferation, migration, and invasion of CRC cells then inhibit the tumor progression of CRC. Miao et al. [66] indicated that circSMARCA5 could significantly promote AT-rich interaction domain 4B (ARID4B) expression via serving as a sponge of miR-39-3p [66]. Furthermore, Yang et al. [67] also found that circSMARCA5 might serve as a ceRNA that binds to microRNA-552, thereby blocking Wnt and YAP1 pathways, and ultimately effectively inhibiting CRC progression. These findings showed that circSMARCA5 provided useful information for the clinical diagnosis and treatment of CRC.

### 2.2. CircSMARCA5 Exerts Cancer-Promoting Functions in a Few Tumors

#### 2.2.1. Osteosarcoma

Osteosarcoma (OS) is a primary malignancy of bone, characterized by malignant mesenchymal cells producing osteoid and/or immature bone, with a high propensity for local invasion and metastasis and a low survival rate [90,91,92]. Zhang et al. [33] identified that circSMARCA5 expression was upregulated in OS, and the overexpression of circSMARCA5 promoted cell proliferation, adhesion, migration, invasion, and OS progression. In addition, to investigate the role of circSMARCA5 in regulating pro-oncogenic mechanisms, a circSMARCA5-mediated ceRNA network was constructed using the CircNet database (http://syslab5.nchu.edu.tw/CircNet/, accessed on 27 September 2020 ), and five miRNAs were screened (miR-17-3p, miR-432-5p, miR-561-3p, miR-10b-3p, and miR-181c-3p) as well as 25 mRNAs interacting with them [33]. However, further experimental validation is required to confirm the precise cancer-promoting mechanism of circSMARCA5 in OS.

#### 2.2.2. Prostate Cancer

Prostate cancer (PCa) is an epithelial malignancy that occurs in the prostate gland and is the third most common cancer in men worldwide [93]. Dong et al. and Kong et al. both identified that the expression of circSMARCA5 was upregulated in PCa cells and tissues [68,69]. Besides this, according to Kong et al. [69], knockdown of circSMARCA5 was able to restrain cell proliferation and growth, limit cell cycle progression, and significantly increase cell apoptosis. Mechanistically, Dong et al. [68] demonstrated that circSMARCA5 could promote cell proliferation, metastasis, and glycolysis in PCa by upregulation of programmed cell death 10 (PDCD10) levels through interaction with miR-432. In summary, the research showed that circSMARCA5 might promote PCa development through the circSMARCA5/miR-432/PDCD10 axis, which might provide an innovative target for clinical diagnosis and treatment of PCa.

#### 2.2.3. Bladder Cancer

Bladder cancer is a heterogeneous and malignant cancer with over 430,000 men and women diagnosed worldwide every year [81,94]. Tan et al. and Zhang et al. [70,71] found increased levels of circSMARCA5 in bladder cancer tissues compared with adjacent tissues. Moreover, Tan et al. [70] demonstrated circSMARCA5 was also overexpressed in bladder cancer cells compared to normal human urothelial cells. Further experiments showed that the overexpression of circSMARCA5 promoted bladder cancer cell proliferation and invasion but repressed apoptosis [70]. In the study by Zhang et al. [71], circSMARCA5 expression was negatively correlated with miR-432 in bladder cancer tissues, but there was no evidence proving the regulatory relationship between them [71]. In addition, high circSMARCA5 expression was correlated with larger tumor size, higher tumor stage, lymph node (LYN) metastasis, shorter disease-free survival (DFS), and overall survival (OS), but low miR-432 expression has the opposite conclusion [71]. Further, multivariate Cox’s regression analysis displayed that high circSMARCA5 expression was an independent predictive factor for both worse DFS and OS in bladder cancer patients [71]. The research suggested that circSMARCA5 might function as a tumor promoter in bladder cancer, and it had the potential to be a promising diagnostic and prognostic biomarker for bladder cancer.

## 3. Molecular Mechanisms of circSMARCA5 Involved in Tumorigenesis and Development

### 3.1. Upstream Regulation Mechanisms of circSMARCA5

The deregulation of circRNA expression in tumors is a critical molecular mechanism that causes carcinogenesis. CircRNA-002178, for example, is up-regulated in LUAD tissues and LUAD cancer cells, which promotes tumor development [95]; while circRAPGEF5 is markedly down-regulated in renal cells and carcinoma tissues, which inhibits tumor growth [96]. CircRNA abundance is regulated by a variety of mechanisms. Intron complementation primitives (ICSs) can boost circRNA expression by promoting reverse splicing of mRNA precursors. RBPs, on the other hand, can stimulate back splicing by directly bridging distal splice sites and regulating back splicing via binding to ICSs [97]. CircSMARCA5 formation is also governed by several regulatory mechanisms (Figure 1). It has been established that the KH domain containing RNA binding (QKI) and DHX9 play a role in this procedure. QKI is an RBP that induces exon cyclization when it binds to the intron QKI binding motif, which in turn promotes the synthesis of circRNA [98]. As circSMARCA5 is formed, QKI binds to SMARCA5 precursor mRNA downstream of exon 15 and upstream of exon 16 to promote circSMARCA5 synthesis in tumors [99]. As a member of a class of nuclear RNA decapping enzymes, DHX9 can inhibit the formation of circRNAs by binding to their flanking reverse complementary sequences and inhibiting Alu element-induced intron pairing [100,101]. I14RC (reverse complementary sequence in intron 14), as well as I16RC (reverse complementary sequence in intron 16) of SMARCA precursor mRNAs, are the targets of DHX9 action, and overexpression of DHX9 suppresses the synthesis of circSMARCA5 [74]. Additionally, the expression level of circSMARCA5 is inevitably influenced by the transcription activity of its parent gene SMARCA5, which promotes the expression abundance of circSMARCA5 and therefore affects tumor development [102].

### 3.2. Downstream Regulation Mechanisms of circSMARCA5

#### 3.2.1. CircSMARCA5 Acts as miRNA Sponges to Promote the Expression of Its Target Genes

The most intensively studied mechanism of circRNA is that involved in the expression regulation of its miRNA target genes as miRNA sponges by competitively binding to miRNAs, thereby interrupting the interaction between miRNAs and their target genes. Tumorigenesis is greatly impacted by the presence of many miRNA response elements (MREs) on circRNA loop sequences, which affect miRNA-induced gene regulation [2,103]. For example, ciRS-7, which contains more than 70 miR-7 binding sites, impacts the development of several illnesses and tumors by preventing miR-7 from performing its biological [104,105] CircTCF25 interacts with miR-103a and miR-107 to promote the proliferation and migration of bladder cancer cells [19]. Circ-ITCH participates in the development of ovarian cancer by sponging miR-145 [106]. Many studies have shown that circSMARCA5 has multiple miRNA binding sites, functions as a miRNA sponge to regulate the expression of its target genes, and is crucial for the development of various tumors (Figure 2A). For example, in gastric cancer, circSMARCA5 acts as a sponge for miR-346 and miR-4295 and inhibits the proliferation, migration, and invasion of gastric cancer cells [63]. CircSMARCA5 could absorb miR-39-3p and miR-552 to inhibit CRC progression [66,67]. By interacting with miR-620, circSMARCA5 inhibits cervical cancer proliferation and invasion [34]. In NSCLC, circSMARCA5 exerts inhibitory effects on its development via the miR-19b-3p/HOXA9 axis [64]. Through the negative regulation of miR-767-5p levels, circSMARCA5 prevents the growth of multiple myeloma cells in MM [58]. In hepatocellular carcinoma, circSMARCA5 inhibits the growth, migration, and invasive ability of hepatocellular carcinoma cells by sponging miR-17-3p, miR-181b-5p, and miR-942-5p [54,55]. In contrast, circSMARCA5 contributes to the development of prostate cancer by sponging miR-432 [68]. In addition, the CircNet database was utilized by the researchers to evaluate five miRNAs that interact with circSMARCA5 in osteosarcoma. These miRNAs include miR-17-3p, miR-432-5p, miR-561-3p, miR-10b-3p, and miR-181c-3p, however additional studies are needed for validation [33]. Since CircSMARCA5 can regulate some miRNAs with tumor-promoting functions, and some other miRNAs with tumor inhibiting functions, this may explain the dual function of CircSMARCA5 in different tumor types.

#### 3.2.2. CircSMARCA5 Binds to RBP Involved in RNA Splicing Regulatory Process

RBPs are a crucial class of proteins that bind to double-stranded or single-stranded RNA. RBPs have RNA-binding domains such as RNA recognition motifs (RRM) and K-homology structural domains (KH) [107], which interact with RNA by recognizing particular RNA-binding domains and are widely involved in several post-transcriptional regulatory processes, such as RNA shearing, translocation, sequence editing, intracellular localization, and translational control [108]. By interacting and binding with diverse RBPs, specialized circRNA protein complexes (circRNPs) can be created that influence the functions of associated proteins, modify cellular biological processes, and participate in the progression of certain malignancies [109]. For example, in NSCLC, circ-UBR5 could bind to the splicing regulatory factor QKI and NOVA alternative splicing regulator 1 (NOVA1) in the nucleus, revealing circ-UBR5 is involved in the RNA splicing regulatory process [110]. It was shown that circSMARCA5 can participate in tumor progression by binding to RBPs at specific binding sites (Figure 2B). In GBM, splicing regulatory factor SRSF1 was found to bind to circSMARCA5 at multiple sites, circSMARCA5 functioned as a tumor suppressor by regulating the activity of SRSF1, thus affecting the splicing and expression of SRSF3 and polypyrimidine tract binding protein 1(PTBP1) and polypyrimidine tract binding protein 2(PTBP2) [60]. In addition, the binding of circSMARCA5 to SRSF1 also regulated the mRNA splicing of VEGFA, thereby preventing tumor angiogenesis, and suppressing the development of glioblastoma [59].

#### 3.2.3. CircSMARCA5 Serves as a Transcriptional Regulator to Regulate Parental Gene Expression

CircRNAs can act as transcription regulators of their parental genes and regulate their transcription activity and expression [111], participating in a variety of biological processes. For example, inhibition of circEIF3J and circPAIP2 reduces the degree of host gene transcription [27]. The circRNA of SEPALLATA3 regulates the splicing of its homologous mRNA through the formation of an R-loop [112]. Ci-ankrd52 acts as a positive regulator of Pol II by accumulating at the ankrd52 transcription site, interacting with the extended Pol II complex, then positively regulating the transcription activity of its parental gene [113]. CircSMARCA5 is negatively correlated with the expression of its parental gene SMARCA5 in breast cancer. According to reports, SMARCA5 is important for maintaining genomic stability as well as controlling DNA damage repair functions [114,115,116]. CircSMARCA5 forms an R-loop by binding to SMARCA5, causing SMARCA5 exon 15 to go into a transcriptional pause, leading to the downregulation of SMARCA5 expression and the production of truncated nonfunctional proteins that exert tumor suppressive effects by inhibiting DNA damage repair function in BC cells [32] (Figure 2C).

## 4. Potential Values of circSMARCA5 in Clinical Diagnosis and Treatment of Cancers

As circRNAs have the characteristics of abnormal stability, conservatism, and tissue-specific expression [28,78,111,117], the differential expression of some circRNAs is strongly correlated with tumor malignancy, clinical progression, and prognosis. Therefore, circRNA has great potential as a diagnostic target for tumors. Many circRNA patents have been used in clinical applications and have played an important role in clinical diagnosis and treatment. For instance, circ-EIF6 can be used to predict the prognosis level of triple-negative breast cancer patients, and its encoded protein EIF6-224aa can be used as a marker and a target for breast cancer prognosis and treatment [118]. For the diagnosis and prognosis of NSCLC, hsa_circRNA_012515 provides a new molecular marker and detection strategy [119]. Through bioinformatic means and studies on clinical cancer patients, it was demonstrated that circSMARCA5 was closely associated with various clinical features, revealing the great potential and value of circSMARCA5 as a diagnostic and prognostic-related biomarker. In addition, circSMARCA5 expression correlated with the degree of tumor differentiation, lymph node metastasis, tumor stage, and TMN stage [34,57,65]. CircSMARCA5 expression levels can be used to predict survival parameters such as overall survival (OS), disease-free survival (DFS), progression-free survival (PFS), and recurrence-free survival (RFS) in patients with tumors [54,58,66]. CircSMARCA5 expression levels can be used to identify patients with hepatocellular carcinoma, cirrhosis, and hepatitis B type hepatocellular carcinoma, and have higher diagnosis efficiency when combined with alpha-fetoprotein (AFP) making it a valid class of biomarkers for the diagnosis of hepatocellular carcinoma [54]. Low expression of circSMARCA5 was an independent adverse predictor of gastric cancer [61], while high circSMARCA5 expression was an independent predictive factor for both worse DFS and OS in bladder cancer patients [71]. Therefore, circSMARCA5 is a promising biomarker for the diagnosis of a series of cancers.

Since the differential expression of circSMARCA5 is strongly associated with tumor progression, accordingly, the intervention of circSMARCA5 expression can reverse the malignant phenotype of tumors and can be used as a tumor therapeutic target. For example, the overexpression of circSMARCA5 could inhibit GC cell proliferation, migration, and invasion [61,63], but promote OS cell proliferation, adhesion, migration, invasion, and OS progression [33]. Moreover, circSMARCA5 was also found to be closely associated with tumor drug resistance. In NSCLC cell lines, overexpression of circSMARCA5 enhanced chemosensitivity to cisplatin and gemcitabine [65]. In BC cells, circSMARCA5 overexpression effectively increased the sensitivity of MCF-7 xenograft tumors to cisplatin treatment [32]. Due to its short sequence and high stability, it is feasible to create circSMARCA5 inhibitors or expression systems to intervene in circSMARCA5 expression for tumor treatment with practical potential. It is also possible to target its upstream and downstream target molecules with gene targeting therapy to achieve tumor growth inhibition, further transforming circSMARCA5 into an effective treatment for tumors.

## 5. Conclusions

In this review, we summarized the current status of circSMARCA5 and its critical role in tumor development and clinical potential. The differential expression of circSMARCA5 in multiple tumors is closely associated with tumor cell proliferation, invasion, metastasis, cell cycle progression, apoptosis, DNA damage repair, angiogenesis, and drug resistance, thus affecting the tumor progression of multiple tumors. CircSMARCA5 has dual roles in tumor development, it plays a tumor-suppressing role in breast cancer, glioblastoma, gastric cancer, multiple myeloma, colorectal cancer, liver cancer, non-small cell lung cancer, and cervical cancer, but exerts a tumor-promoting role in osteosarcoma, prostate cancer, and bladder cancer. Increasing evidence shows that tumor is a metabolism-related disease, in which glycolysis is an important way for tumor cells to obtain energy and intermediate products, thus maintaining their malignant phenotype. CircSMARCA5 promotes the expression and activity of LDHA and lactate production of tumor cells in prostate cancer [69], while it inhibits the uptake of glucose and glycolysis in liver cancer and gastric cancer [55,62]. Therefore, circSMARCA5 plays a dual role in promoting or inhibiting the glycolysis of tumors, thus it plays critical roles in tumorigenesis and tumor progression as an oncogene or tumor suppressor. Mechanically, the expression level of circSMARCA5 is regulated by QKI as well as DHX9. CircSMARCA5 plays a critical role in tumor progression and drug resistance by acting as miRNA sponges to increase the expression of corresponding downstream target genes as well as regulating gene splicing and expression of its target genes including its host gene. Based on the stability, expression specificity, and function of circSMARCA5 in tumors, it can become a potential diagnostic and therapeutic target for different types of tumors.

## Figures and Tables

**Figure 1 cells-11-03074-f001:**
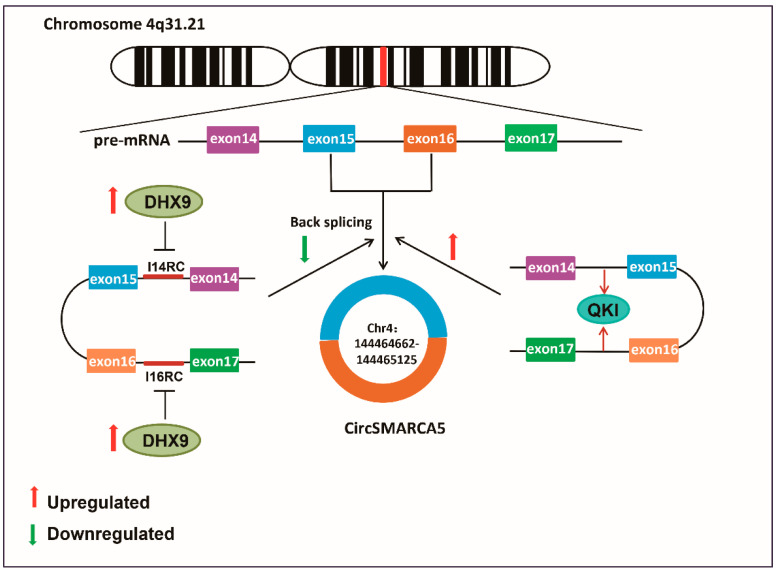
Biogenesis and upstream regulation mechanisms of circSMARCA5. CircSMARCA5 is located on Chromosome 4q31.21 and derived from exon 15 and exon 16 of the SMARCA5 gene, containing 464 nucleotides (chr4:144464662-144465125). I14RC (the reverse complement sequence in intron 14) and I16RC (the reverse complement sequence in intron 16) of the mRNA of SMARCA5 precursor have the action target of ExH-box helicase 9(DHX9), and the upregulation of DHX9 inhibits the production of circSMARCA5. KH domain containing RNA binding (QKI) combines upstream and downstream exon 15 and exon 16 at the site of circSMARCA5 formation on the mRNA of the SMARCA5 precursor to promote circSMARCA5 formation in the tumor.

**Figure 2 cells-11-03074-f002:**
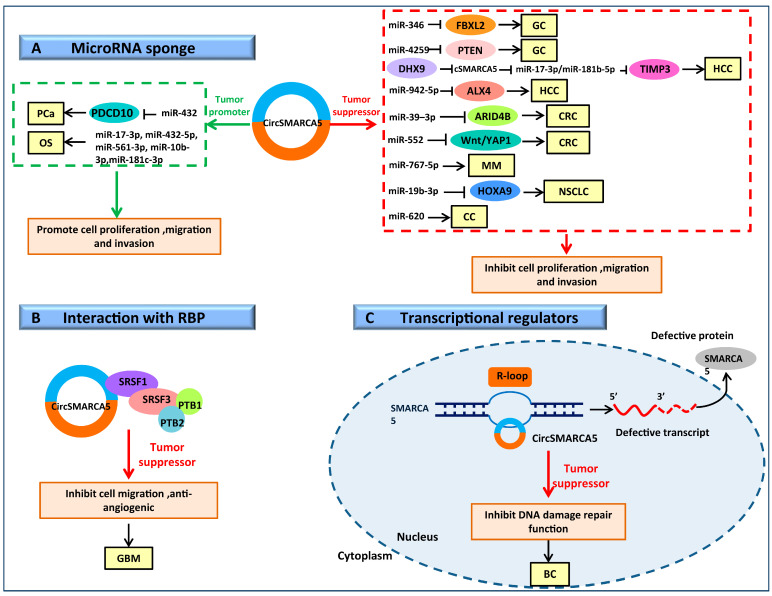
Mechanisms in various types of tumors of circSMARCA5. (**A**) CircSMARCA5 acts as a miRNA sponge. CircSMARCA5 functions as a miRNA sponge to adsorb multiple miRNAs, thus regulating the expression of downstream genes and cancer progression. (**B**) CircSMARCA5 interacts with RNA binding protein SRSF1 thus regulating the cell migration and angiogenic potential. (**C**) CircSMARCA5 can bind to SMARCA5, forming an R-loop, resulting in the production of a truncated nonfunctional protein, thus inhibiting DNA damage repair function.

**Table 1 cells-11-03074-t001:** Functional characteristics of circSMARCA5 in multiple human cancers.

Cancer Types	Expression in Cancers	Roles	Regulatory Mechanism	Function Roles	References
Hepatocellular carcinoma	Down	Tumor suppressor	miR-17-3p/miR-181b-5p-TIMP3 miR-942-5p/ALX4	Promote apoptosis. Inhibit cell proliferation, migration, invasion, epithelial-mesenchymal transition, cell cycle, and glycolysis.	[53,54,55,56]
Intrahepatic cholangiocarcinoma	Down	Tumor suppressor	N/A	Inhibit cell proliferation.	[57]
Multiple myeloma	Down	Tumor suppressor	miR-767-5p	Promote apoptosis. Inhibit cell proliferation arrest.	[58]
Cervical Cancer	Down	Tumor suppressor	miR-620	Inhibit cell proliferation, migration, and invasion. Induced cell cycle arrest.	[34]
Glioblastoma	Down	Tumor suppressor	SRSF1/SRSF3/PTB	Inhibit cell migration, anti-angiogenic.	[59,60]
Breast Cancer	Down	Tumor suppressor	SMARCA5	Inhibit DNA damage repair function.	[32]
Gastric Cancer	Down	Tumor suppressor	miR-346/FBXL2 miR-4295/PTEN	Inhibit cell proliferation, migration, invasion, and glycolysis.	[61,62,63]
Non-Small Cell Lung Cancer	Down	Tumor suppressor	miR-19b-3p/HOXA9	Inhibit cell proliferation, migration, and invasion.	[64,65]
Colorectal Cancer	Down	Tumor suppressor	miR-39-3p/ARID4B miR-522/Wnt/YAP1	Inhibit cell proliferation, migration, and invasion.	[66,67]
Osteosarcoma	Up	Tumor promoter	miR-17-3p, miR-432-5p, miR-561-3p, miR-10b-3p, miR-181c-3p	Promote cell proliferation, migration, and invasion.	[33]
Prostate cancer	Up	Tumor promoter	miR-432/PDCD10	Promote cell proliferation, migration, glycolysis, and cell cycle. Inhibit apoptosis.	[68,69]
Bladder cancer	Up	Tumor promoter	N/A	Promote cell proliferation and invasion. Inhibit apoptosis.	[70,71]

## Data Availability

Not applicable.

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
