# Peer review of "The Emerging Roles and Clinical Potential of circSMARCA5 in Cancer"

_cells, 2022, doi:10.3390/cells11193074_

Round 1
Reviewer 1 Report
In this review article by Xue at al. the authors give a complete overview of recent research on expression of circSMARCA5 (hsa-circ-0001445) in various types of cancer. They cover findings of tumor-suppressive and oncogenic function of circSMARCA5 in cancers, describe the mechanisms of its molecular regulation, then discuss circSMARCA5’s ability as a biomarker for cancer diagnosis and a potential approach for gene targeted therapy. This is a nice summary that will help readers gain a more complete knowledge of the role of circSMARCA5 in cancer cell abnormality, and its possibility for clinical application.
The specific comments that need to be addressed are as follows:
1. The cancer types summarized in Table 1 are not corresponding well with their description in the text, which confuses the reader. The authors should keep the order and terminology of cancer types described in the text consistent with Table 1.
2. The authors should include circSMARCA5 expression and its role in bladder cancer in the article, as it is reported in publications:
- Tan Y, Zhang T, Liang C. Circular RNA SMARCA5 is overexpressed and promotes cell proliferation, migration as well as invasion while inhibits cell apoptosis in bladder cancer. Transl Cancer Res. 2019 Sep;8(5):1663-1671.
- Zhang Z, Sang Y, Liu Z, Shao J. Negative Correlation Between Circular RNA SMARC5 and MicroRNA 432, and Their Clinical Implications in Bladder Cancer Patients. Technol Cancer Res Treat. 2021 Jan-Dec;20:15330338211039110.
3. Also, the review would benefit with an update from the studies listed below:
- Xu Q, Zhou L, Yang G, Meng F, Wan Y, Wang L, Zhang L. Overexpression of circ_0001445 decelerates hepatocellular carcinoma progression by regulating miR-942-5p/ALX4 axis. Biotechnol Lett. 2020 Dec;42(12):2735-2747.
- Yang S, Gao S, Liu T, Liu J, Zheng X, Li Z. Circular RNA SMARCA5 functions as an anti-tumor candidate in colon cancer by sponging microRNA-552. Cell Cycle. 2021 Apr;20(7):689-701.
4. miR-39-3p from the Table 1 is incorrectly misspelled as miR-93-3p in the text (line 198).
5. The text needs some revision in spelling and grammar.
Author Response
Response to Reviewer 1 Comments
1.The cancer types summarized in Table 1 are not corresponding well with their description in the text, which confuses the reader. The authors should keep the order and terminology of cancer types described in the text consistent with Table
R: Thanks a lot for the reviewer’s helpful comment and reminder. We have corrected cancer types summarized in Table 1, and the order and terminology of the cancer types shown in Table 1 are now consistent with the text (Page 3, line 80 and 81).
2.The authors should include circSMARCA5 expression and its role in bladder cancer in the article, as it is reported in publications:
-Tan Y, Zhang T, Liang C. Circular RNA SMARCA5 is overexpressed and promotes cell proliferation, migration as well as invasion while inhibits cell apoptosis in bladder cancer. Transl Cancer Res. 2019 Sep;8(5):1663-1671.
-Zhang Z, Sang Y, Liu Z, Shao J. Negative Correlation Between Circular RNA SMARC5 and MicroRNA 432, and Their Clinical Implications in Bladder Cancer Patients. Technol Cancer Res Treat. 2021 Jan-Dec; 20:15330338211039110.
R: Many thanks for the reviewer’s valuable suggestion that will make our our manuscript more comprehensive. According to the references provided by the reviewer and other relevant reports of this subject, we have summaried the function and mechanism of circSMARCA5 in bladder cancer (Page 7, line 251 -268).
3.Also, the review would benefit with an update from the studies listed below:
-Xu Q, Zhou L, Yang G, Meng F, Wan Y, Wang L, Zhang L. Overexpression of circ_0001445 decelerates hepatocellular carcinoma progression by regulating miR-942-5p/ALX4 axis. Biotechnol Lett. 2020 Dec;42(12):2735-2747.
-Yang S, Gao S, Liu T, Liu J, Zheng X, Li Z. Circular RNA SMARCA5 functions as an anti-tumor candidate in colon cancer by sponging microRNA-552. Cell Cycle. 2021 Apr;20(7):689-701.
R: Thank a lot for the reviewer’s valuable comments. These information have been added to the manuscript under " The dual functions of circSMARCA5 in tumors" (Page 4, line93-96 and 105-108; Page 6, line219-221) as the reviewer suggested, and we believe that the changes would make our manuscript more comprehensive.
4.miR-39-3p from the Table 1 is incorrectly misspelled as miR-93-3p in the text (line 198).
R: We appreciate the reviewer’s suggestion and sorry for the carelessness, we have corrected the error in text as indicated (Page 6, line 219).
5.The text needs some revision in spelling and grammar.
R: Thanks a lot for the reviewer’s suggestion and reminder. The manuscript has been further revised by a native English speaker to improve the grammar and readability as suggested.
Reviewer 2 Report
The review article: The emerging roles and clinical potential of circSMARCA5 in cancer
The authors highlighted an inclusive role of circSMARCA5 in different types of cancer as well as its gene regulation. The review provides detailed interaction of circSMARCA5 with other regulatory molecules which can help researchers to understand the big picture of the dynamics of its role in cancer.
The review can be published after these minor modifications:
Line 59: explain what is DHX9 in brief
Line 60: explain QKI
Line 65: mention, in brief, the dual function of circFOXO3 and circ-CDR1as. Also, be consistent in naming circRNAs
Line 85: remove the full stop after ref [62,63,68]. Same for line 88, 97, 104, 117, 128, 182, 194, 205
Line 92, mention the main function for miR-17-3p and -181b-5p
Line 91-96: long sentence. need to be rephrased
Author Response
Response to Reviewer 2 Comments
1.Line 59: explain what is DHX9 in brief
R: Thanks a lot for the reviewer’s comment and reminder. We have provided a more specific description for DHX9 in the manuscript (Page 2, line 59).
2.Line 60: explain QKI
R: We thank the reviewer for pointing out this issue. We have added the explanation of QKI as required (Page 2, line 60).
3.Line 65: mention, in brief, the dual function of circFOXO3 and circ-CDR1as. Also, be consistent in naming circRNAs
R: We acknowledge the reviewer’s comment very much, which are valuable in improving the quality of our manuscript. We have briefly provided the description of the dual roles played by circFOXO3 and circ-CDR1as in tumors (Page 2, line 65-68).
4.Line 85: remove the full stop after ref [62,63,68]. Same for line 88, 97, 104, 117, 128, 182, 194, 205
R: We are extremely grateful to the reviewer for pointing out this problem. We have removed the full stop after ref [62, 63, 68], and we have also revised the errors in oher places with the same problem in the text.
5.Line 92, mention the main function for miR-17-3p and -181b-5p
R: It is a pretty good suggestion, thanks a lot. We have provided a specific description for the main function of miR-17-3p and -181b-5p involved in HCC (Page 4, line 100-101).
6.Line 91-96: long sentence. need to be rephrased
R: Thanks a lot for your suggestion. We have carefully checked and improved the English writing in the revised manuscript (Page 4, line 97-105).